# Anxious Personality Traits: Perspectives from Basic Emotions and Neurotransmitters

**DOI:** 10.3390/brainsci12091141

**Published:** 2022-08-27

**Authors:** Jie Dong, Tingwei Xiao, Qiuyue Xu, Fei Liang, Simeng Gu, Fushun Wang, Jason H. Huang

**Affiliations:** 1Institute of Brain and Psychological Sciences, Sichuan Normal University, Chengdu 610066, China; 2School of Nursing, Nanjing University of Chinese Medicine, Nanjing 210023, China; 3Department of Psychology, Medical School, Jiangsu University, Zhenjiang 210023, China; 4Department of Neurosurgery, Baylor Scott & White Health, Temple, TX 79409, USA; 5Department of Surgery, College of Medicine, Texas A&M University, Temple, TX 79409, USA

**Keywords:** personality traits, anxiety, neurotransmitter, emotion

## Abstract

Objective: Recently, many emotional diseases, such as anxiety and depression, have prevailed, and it is expected that emotional disease will be the leading cause of social and economic burden in 2030. These emotional diseases may be due to certain personality traits, which could be the reasons for the development of mental illness. Personality theories have been constantly developed over the past hundreds of years, and different dimensions of personality traits corresponding to different physiological bases and emotional feelings have been proposed. However, personality may be the least studied area in psychology. Methods: In this paper, we will give a short review on the development of personality theories as well as dimensional emotional theory. Then, we will compare the similarities between the emotional dimension and personality dimension. Furthermore, we will also investigate the neural mechanisms of personality and emotions, focusing on neuromodulators for anxiety-related personality traits, in order to provide a clear relationship between different neurotransmitters and anxiety-related personality traits. Results: The results of our study suggest that the emotional dimension and personality dimension may be somewhat related, for example, the extrovert/introvert dimension of personality might be related to the hedonic dimension, which includes happiness/sadness, and the neurotic dimensions might be related to emotional arousal. In addition, our study found that personality traits are also related to basic emotions, for instance, people who are too self-centered are prone to feeling a mood of disgust or depression, while anger and fear correspond to unstable personality traits. The analysis suggested that the neural substrates of both personality and emotions might be described as follows: extroverted–joy–dopamine (DA); introverted–disgust–5-hydroxytryptamine (5-HT); unstable (neuroticism)–anger/fear–noradrenaline (NE); stable–calmness. Conclusions: The results of this study suggest that there is a correlation between personality traits and emotions, and both depend on monoamine neurotransmitters (dopamine, norepinephrine and serotonin). In addition, personality disorders can be interfered via the regulation of emotions and neurotransmitters. This paper opens up a whole new perspective for future research on personality traits and emotional diseases and has great clinical value and practical significance.

## 1. Introduction

Anxiety disorders are a group of mental disorders that are manifested clinically as anxiety syndrome. Individuals suffering from an anxiety disorder excessively avoid and fear perceived threats in the environment (such as social occasions or unfamiliar places) or from their own mind (such as unusual physical feelings). Anxiety disorders are a complication of major depressive disorders, alcoholism, substance use and personality disorders [1,2]. In addition, it has been suggested that up to 20% of cancer patients suffer from depression and 10% from anxiety disorders, compared to 5% and 7% of the general population, respectively [3].

A high risk of developing anxiety disorders is associated with the anxious personality trait [4,5], which is a kind of stable and fundamental behavioral tendency supported by biological mechanisms [6,7]. In Western culture, the biological mechanisms of personality have long been investigated [8]. Hippocrates first proposed the “humoral doctrine”, which ascribed different temperament types to four kinds of body fluids. Recently, Cloninger proposed a physiological theory of personality, which suggests that the three monoamine neurotransmitters correspond to the three dimensions of personality: dopamine corresponds to reward dependence, 5-hydroxytryptamine corresponds to punishment, and norepinephrine corresponds to novelty seeking [9]. Although many studies have tried to determine the physiological factors of personality traits, there is not a well-accepted theory about personality or a direct link between neurotransmitters and personality traits [10,11].

Emotion is a very important aspect of personality, and emotions have a similar dimensional construction [12,13]. The dimensional construction theory suggests that emotions are constructed in two dimensions (arousal and valence), and all emotions can be located on the quadrant constructed with two major axes: arousal and valence [14]. However, personality is much more complicated than emotions; thus, emotional studies might help to shed some light in personality studies. Here, we will attempt to compare recent emotional studies with personality traits to systematically and comprehensively explore personality traits from the perspective of emotions and try to relate them to chemicals in the body [15]. Finally, we will attempt to connect personality with chemicals in the body by putting forward the following correspondence (personality trait–emotion–neurotransmitter) [16].

Certain personality traits are related to certain psychiatric problems [17], and in-depth studies of personality traits could prevent psychiatric disorders. Many studies have found that certain personality traits are related to certain chemicals in the body. For example, Hippocrates theorized that personality traits and human behaviors were based on the four kinds of blood fluids. Lester proposed that psychoticism, neuroticism and extraversion were associated with different levels of dopamine, serotonin and norepinephrine, respectively [18]. Other studies have shown that personality characteristics are associated with brain domains, for example, in phrenology [15]. Recently, one study tried to interpret personality traits from the perspective of EEG (electroencephalogram), and it was found that people who had slow and weak EEG recordings tended to develop extraverted behavior patterns, while people with fast and relatively strong EEG recordings easily formed an introverted behavior pattern [19]. Interestingly, Cahill and Polich found that P300 amplitudes were related to personality type, with introverts generally having smaller amplitudes than extroverts [20].

In addition to physiological perspectives, numerous studies have examined the relationship between personality traits and emotional states or mental disorders. Magee and Biesanz examined the ways in which the Big Five personality states affected short-term experiences of well-being, and the results showed that emotional stability and extraversion exhibited the strongest associations with well-being, followed by conscientiousness and then agreeableness [21]. In addition, another group studied the “Big Five” personality traits and their relationship to the daily experiences and regulation of anger; however, they found that personality traits offered minimal value in predicting anger in daily life [22]. The correlation between personality traits and mental disorders is also a research hotspot. Neuroticism has been found to be significantly associated with major depression, anxiety disorder, substance use disorder (SUD) and psychopathological severity [23]. Other studies have suggested that neuroticism, psychoticism, impulsivity and aggression are risk factors for suicidal ideation, while extraversion has a protective effect against suicidal ideation [24].

Currently, few studies have systematically connected personality traits directly with emotions and neurotransmitters, and personality might be the least studied area in psychology. So, what is the connection between personality traits and emotions? In addition, what are the neural or humoral mechanisms corresponding to personality traits? To identify the relationship between them and emotional manifestations and the neural or humoral mechanisms of anxiety as a personality trait, we analyzed personality traits from the perspective of emotions and neurotransmitters, while focusing on the emotional manifestations and neuro-humoral mechanisms of anxiety-related personality traits in a systematic way. We hope this paper will provide a new understanding and novel perspective of the neurotransmitter basis for personality traits, which might shed light on the development of neuro-pharmaceuticals.

## 2. Historical Development of Personality Theories

Personality refers to the long-term traits that propel individuals to think, feel and behave consistently in specific ways. The study of personality can be traced back 2000 years to Hippocrates [25], who theorized that personality was based on four kinds of body fluids: melancholic temperament was due to black bile from the kidneys, sanguine temperament as due to red blood from the heart, choleric temperament was due to yellow bile from the liver and phlegmatic temperament was due to white phlegm from the lungs [26,27,28,29,30]. At the same time as Hippocrates, Huangdi (*The Inner Canon of Huangdi*) also proposed a similar emotional analysis: kidney–fear, heart–joy, liver–anger, lung–sadness, and spleen–missing. Later, Galen extended Hippocrates’ s theory and detailed the four kinds of personality as follows: the melancholic person is sad and reserved and, in contrast, the sanguine person is happy and joyful, while the choleric person is bold and ambitious, and the phlegmatic person is calm and thoughtful [26,31]. Later on, many other researchers refined the four primary temperament types, including Immanuel Kant (in the 18th century) and psychologist Wilhelm Wundt (in the 19th century). Kant agreed with the four-temperament theory of Galen and suggested there was no overlap between the four categories [30,32]. In addition, Kant developed a list of traits for each of the four temperaments [30,31,32,33]. This humor theory was prevalent for over 2000 years and continued to be popular until now in both Western and Eastern culture.

The theory of personality traits has been continuously developed and refined. Wundt was the first to suggest that personality could be better described using two major axes: emotional/non-emotional and changeable/unchangeable [33]. Eysenck refined the theory and proposed the two dimensions as: extroversion/introversion and emotion (Figure 1) [34]. Afterwards, Eysenck renamed the two personality dimensions as neuroticism and extroversion, which were identified as major components in psychological tests [35], and developed the Eysenck Personality Questionnaire (EPQ). Thus, Eysenck viewed people as having two specific personality dimensions: extroversion/introversion and neuroticism/stability (Figure 1) [34,36,37].

However, other studies have suggested different dimensions, for example, Sigmund Freud’s psychodynamic perspective of personality was the first comprehensive theory of personality, explaining a wide variety of both normal and abnormal behaviors [30]. According to Freud, unconscious drives are influenced by a kind of body fluid, libido, and are the forces for our personality. Costa and McCrae developed the famous NEO Personality Questionnaire and suggested five personality factors [39]. The five-factor model of personality is a hierarchical organization of personality traits in terms of five basic dimensions: extroversion, agreeableness, conscientiousness, neuroticism and openness to experience [35]. Lester suggested that psychoticism, neuroticism and extroversion are associated with different levels of dopamine, serotonin and norepinephrine, respectively [18]. In all, personality studies are quite complicated, and most studies have tried to connect temperament to body fluid in order to make it simple. Thus, in this paper, we are trying to suggest that it is similar for emotional dimension and personality dimensions, in that they both share similar neurotransmitters (Figure 1).

## 3. The Similarity between Eysenck’s Personality Dimension and Emotional Dimensions

In this paper, we will continue to propose the four kinds of personality and advance them with the emotional dimensions; in addition, we suggest that the arousal dimension is similar to the personality emotional dimension, and the hedonic dimension might be related to the extrovert dimension (Figure 1). Extroversion and introversion are similar to the hedonic dimension in that they show similar behavioral directions (approaching = extrovert; avoidance = introvert) [40]. In addition, we also suggest that the dimensions for both personality and emotions depend on neurotransmitters. These two dimensions are described below.

### 3.1. Extroversion/Introversion

Extroversion refers to feeling comfortable and enjoyable in the company of others. Extroversion and introversion were most clearly proposed by Carl Jung, who suggested that an extroverted person is energized by being outgoing and socially oriented, while an introvert person is quiet and reserved [41]. These ideas are considered as Jung’s most important contributions to the field of personality psychology, as almost all models of personality now include these concepts [42,43]. Extroversion and the hedonic value are similar in that they both take an approaching manner towards the environment; consistent adjectives for extroversion include active, assertive, energetic, enthusiastic, outgoing and talkative [35]. In contrast, introverts like to take an avoidant behavior towards the environment and are described as quiet, reserved and insensitive to their environment [44]. In addition, introverted personalities are similar to sad emotions in the hedonic dimension (Figure 1), in that they are relatively sensitive to punishment and frustrated non-reward stimulations than extroverted personalities [45]. There may be no goodness or badness associated with being an extrovert or introvert, but some studies suggest that extroversion is associated with positive emotions, whereas introversion is associated with negative emotions [46].

In addition, Horney’s theories also related extroversion/introversion to unconscious anxiety by suggesting that there are three types of coping styles related to anxiety [47,48]. The first is moving towards the environment (people), which suggests attachment, love, acceptance, dependence and relief from anxiety [49]. The second coping style is characterized by moving against people and relies on aggression and assertiveness. The third coping style is moving away from people, and these people handle their anxiety by withdrawing from the world, detachment and isolation; they like privacy and avoid such things as love and friendship, and they also tend to gravitate toward careers that require little interaction with others [48].

### 3.2. Neuroticism (Stable/Unstable)

The vertical dimension was suggested by Wundt to be strong emotion/low emotion, Eysenck named it as the stable/unstable dimension, and it is similar to neuroticism in the Big Five Model. Emotional instability, which is the typical feature of neuroticism, is the personality trait most closely associated with mental health challenges [50]. It is associated with norepinephrine and the sympathetic nervous system, which regulate emotional arousal and control sympathetic autonomic responses [43]. Individual differences in emotional stability are thought to reflect the core of neuroticism. However, this was questioned by Kalokerinos, because it does not reflect the average level of negative emotions [50]. Based on the current definition of neuroticism, it is not limited to refer to high levels of negative emotions, but rather to high reactivity to emotional fluctuations, i.e., emotional variability, which is why some scholars have used “emotional stability” as the opposite of neuroticism in their research [51]. Considering the core definition of neuroticism and previous research, we suggest that the two personality traits of “neuroticism” in the FFM model and “emotional stability” in the “Big five” are the same dimension. Tong used an ecological momentary assessment method among undergraduate students who were asked to display their negative emotions and assessments in a natural environment for two consecutive days at regular intervals [52]. The results showed that individuals with high neuroticism showed more negative appraisal styles than those with low neuroticism, and more importantly, the higher the degree of neuroticism, the stronger the appraisal–emotion relationship for the four negative emotions (anger, sadness, fear and guilt).

Some studies have attributed anxiety to a sub-dimension of neuroticism in the Big Five model [39,53]. In addition, studies have shown that the percentages of variance explained by the trait component for the anxiety and personality constructs (73–84%) are significantly greater than those explained by the trait component for depression (46%) [54]. These findings suggest that the symptoms of anxiety are more similar to personality variables.

## 4. The Similarity between Personality Traits and Basic Emotions

A great deal of evidence suggests that personality and emotions are linked via coping behaviors and that emotional features can be viewed as an emergent attribute for personality [55]. Personality traits are the building blocks of personality, and psychologists have made many efforts to define and organize personality traits. They are often bundled together based on broad personality factors, such as the dimensions or the Big Five trait taxonomy. These personality traits are closely related to emotional characteristics and reflect emotional types and intensity in certain situations, as well as how individuals learn to cope with these emotions [56]. Personality traits can in turn influence how people feel emotionally when things come up. However, personality can be sliced in many different ways, and different personality traits are associated with different features of emotions. As we previously suggested, two strong predictors of emotional experience are extroversion and neuroticism: extroversion is associated with positive emotions and neuroticism is associated with instable emotions [57]. Here, we will try to see the relationship between personality traits and basic emotions, which are the building blocks of all emotions.

### 4.1. Basic Emotional Theory and Personality Traits

Basic emotional theory, which suggests that there exists a limited number of basic emotions, has been a prevalent theory in past decades [58,59,60,61]. The basic emotions were evolved to handle fundamental life tasks, such as finding something to eat (joy), avoiding being eaten (fear) or avoiding toxic food (disgust) [62]. They can activate the body to deal with prototypical situations that have significant implications for survival. In addition, they are fairly common to all animal species in their interactions with their external environment, conspecifics and members of other species [61]. However, these basic emotions cannot be broken down further into more basic psychological components. In recent years, basic emotion theory has stimulated a number of empirical studies. Thus far, the accepted basic emotions are *fear, anger, joy, sadness and disgust* [14,63]. However, some psychologists are still debating whether to choose dimensional theory or basic emotional theory.

In our previous paper, we integrated basic emotion theory with dimensional emotions and suggested that basic emotion theory and dimensional theory are not contrary to each other [64]. The integrative approach proposes that “basic emotions”, like all emotions, are also constructed by “core affects”, and their locations can also be found on the circumplex [38]. Therefore, the integrative theory proposes that: The reason for the basic emotions to be “basic” is that they are located on the axis on the circumplex (dimension). Happiness resides on the positive pole of the hedonic dimension, sadness is found on the negative pole of the hedonic dimension, and fear and anger reside at the top of the arousal axis (Figure 1). Next, we will try to investigate the relationships between basic emotions and personality traits.

### 4.2. Extroverted—Joy

Both extroverted personalities and joy represent the positive pole of the horizontal dimension (Figure 1). So, is there any connection between extroversion and joy? Indeed, it has been found that extroverts are happier than introverts, as has been demonstrated by many reliable observations in the literature [65]. This might be due the fact that all of the things that satisfy our needs are from the outside, and we have to be extroverted to get what we need. Indeed, Canli suggested that brain activation in positive emotion scenarios was related to extroversion traits, and their experimental results showed that brain activation for the happy emotion was significantly and positively correlated with the degree of extroversion and that extroversion was not significantly correlated with activation for other emotions (anger, fear and sadness) [66]. The correlation between personality traits and happiness suggests that certain types of people tend to experience higher levels of happiness than others [21], and that positive emotions are an important component of happiness. Spinhoven studied whether extroversion could predict future happiness and emotional well-being [67]. It was found that extroversion was positively correlated with well-being, and extroversion had a unique contribution to the overall emotional evaluation of happiness and emotional well-being, exceeding the influence of emotional disorders or the severity of their symptoms. It was further suggested that hope and social support play a mediating role between extroversion and well-being [68]; specifically, the mediating role of social support suggests that extroverts are more likely to believe that they can get help from family, friends and others when necessary, and that their subjective perceptions of social support in turn increase their well-being. However, it has also been suggested that the link between extroversion and well-being should be attributed to energy levels and cannot be generalized to the trait level as a whole; energy levels include items reflecting general energy (“full of energy” and “less active than others”), as well as items related to positive expectations (“shows great enthusiasm” and “rarely feels excited or eager”) [69].

In conclusion, extroverted personalities are more likely to feel happy emotions and thus have a strong sense of well-being; however, it has also been suggested that the association between extroverted personality traits and happiness cannot be attributed solely to the level of the trait as a whole, and that this link needs to be further studied at different levels.

### 4.3. Introverted—Disgust (Sadness)

Contrary to extroversion and happiness on the right side of the hedonic dimension, introversion and disgust (sad) are found on the left side of the hedonic dimension. Is there a correlation between introversion and disgust? Indeed, it has been found that people who are overly egotistical and introverted are prone to becoming disgusted or depressed. In their experiment, Clarke found that subjects reported experiencing a generalized, persistent sense of self-loathing, which became more intense when it was necessary to focus on an aspect of oneself, as well as severe psychological and behavioral reactions to self-loathing [70]. Introversion is a major and important core personality trait in patients with major depression [71], and disgust is also strongly associated with depression; even self-loathing predicts suicide risk in PTSD patients [72]. The personality trait of introversion focuses attention to oneself and thus increases the likelihood of becoming disgusted, as disgust occurs due to our own excretions (such as feces or saliva). If we are introverted, we are more likely to feel these excretions and more likely to be disgusted. Thus, the hedonic axis also represents openness to the outside, extroversion, or an approaching manner.

Currently, most studies focus on extroverted personality traits and fewer studies have been conducted on introverted personality traits, and even fewer have explored their correlation with disgust. Future research could focus on whether introverted personalities are more likely to develop disgust or sadness, which can lead to psychiatric disorders such as depression and anxiety. Both disgust and sadness are basic emotions and are located on the left pole of the hedonic axis; however, few studies have probed the difference between them. It might be that disgust is related to disliked things, while sadness is related to the loss of loved ones. Many studies have probed the role of sadness in depression, while disgust as a basic emotion has been investigated in few studies [14]. However, it was recently found to be the major basic emotion related to most mental disorders, such as Obsessive-compulsive disorder, anxiety or depression [73]. There is increasing evidence implicating disgust in the etiology and maintenance of various types of anxiety disorders [74,75,76]. Consistent with this, Bosman suggested that disgust often complements fear as a common feature of specific phobias, such as spider phobia, and a contamination-based obsessive-compulsive disorders [77].

### 4.4. Neuroticism (Instability)—Anger/Fear

Neuroticism represents emotional instability or emotional arousal, which are located on the top of the vertical dimension. It has been found that neurotic individuals—who have low activation thresholds, cannot inhibit or control their emotional responses and experience high emotional arousal (fight or flight, or fear and anger emotions) in the face of small stressors—are easily stressed or frustrated [78]. Fight or flight corresponds to the emotions of anger and fear [64], and it was shown in a study conducted on Turkish university students that high neuroticism was only associated with “fear/anxiety”. Freud referred to the tendency to frequently respond to fear or experience generalized anxiety as neuroticism [79]. Park found that a highly neurotic person is more likely to experience fear and anger [80]. Moshirian Farahi explored the relationship between neuroticism and emotional face processing valence in adolescents [81]. A relationship between neuroticism and fearful emotions was shown and the interaction of neuroticism and mid-frontal EEG asymmetry significantly affected fear valence. In addition to studies with adolescents, cross-sectional studies have been carried out with women over the age of 70 on whether the core personality trait dimension of neuroticism predicts a fear of falling, showing that women with higher neuroticism scores are more likely to experience a fear of falling [82].

In our previous studies, we suggested that the vertical dimension, called arousal, is due to un-expectancy and affects our safety needs [64]. We proposed the that two dimensions of emotions represent physiological needs (hedonic value) and safety needs (arousal). Everything that happens to us has two features: (a) whether it fits our physiological needs and (b) whether it was expected [64]. If it was expected we feel safe; if it was not expected, we feel threatened and the sympathetic response is activated, which is called “fight-or-flight” behavior. Fight or flight is actually fear and anger emotions.

The reason for associating both fear and anger with the dimension of neuroticism (emotional instability) (Figure 1) is that anger and fear (fight or flight) are perhaps the same emotion, like two sides of the same coin; thus, anger and fear can be placed in the same position on the axis [38]. However, it has also been found that broad personality traits (i.e., the “Big Five”) are less related to angry experiences in everyday life [22]. The correspondence between neuroticism and anger needs to be further explored. Nevertheless, it is certain that neuroticism is associated with fearful emotions. Thus personality can be divided into three kinds: stressful-neuroticism-easy to have fearful or anxious emotions; joy-ful-extroverted and relaxed-easy to be happy and agreeable; depressed-introverted and conscious-easy to be sad and disgusted at the things around.

## 5. Neurotransmitters Subside with Personality Traits

For the past century, monoamine neurotransmitters have been the neural basis for emotions, and first-line treatments for depression are still based on monoamine neurotransmitters. Previously, we suggested that monoamine neurotransmitters may be the neural basis for separate basic emotions: dopamine–joy; norepinephrine–fear (anger); and serotonin–disgust (sadness) [83]. What role do neurotransmitters play in personality traits? Past studies have proposed a kind of relationship between personality traits and neurotransmitters, such as the physiological theory of personality proposed by Cloninger [9], who stated that these neurotransmitters correspond to the three dimensions of personality: dopamine and novelty seeking, 5-hydroxytryptamine and harm avoidance, and norepinephrine and reward dependence. Lester suggested that psychoticism, neuroticism and extroversion are associated with different levels of dopamine, serotonin and norepinephrine, respectively [18]. With the development of modern research, a different perspective on the correspondence between neurotransmitters and personality dimensions will be presented in this paper. The following describes the possible relationship between personality traits and neurotransmitters by combining the two dimensions of Eysenck’s personality theory: extroversion (extroverted/introverted) and neuroticism.

### 5.1. Extroverted—Dopamine (DA)

Dopamine (DA) is a rewarding neurotransmitter found in the brain, and it is currently a synonym for joy or happiness. Indeed, extroversion is also associated with the dopaminergic system [84]. Through their research, Smillie and his colleagues found that extroverts show greater emotional responses only when they respond to reward-seeking situations (e.g., apparent appetitive stimuli) [65]. It is worth noting that dopamine and rewarding behavior are closely linked, and it is known that extroverted personality traits are also inextricably linked to dopamine. King studied cerebrospinal fluid (CSF) dopamine levels in 16 male patients and also administered the extroversion test from the Eysenck Personality Inventory, which showed that CSF dopamine levels were significantly associated with extroversion [85]. In addition, it has also been hypothesized, based on extensive animal studies, that extroversion may be associated with increased dopamine release [86]. The interaction between the population genetics of dopamine and climate was supported by Fischer and Verzijden [87], who noted that individuals from populations with highly efficient dopamine systems tend to favor behavioral style traits like extroversion and emotional stability due to higher perceived reward values, where higher dopamine index scores are associated with higher levels of extroversion; however, this only applies to climatically demanding environments.

In all, some works have shown that extroversion is associated with reward, and dopamine is strongly associated with rewarding behavior. In addition, there is also direct research showing that dopamine is associated with extroversion.

### 5.2. Introverted—5-Hydroxytryptamine (5-HT)

The role of 5-HT in basic emotions is quite controversial, because 5-HT has 14 kinds of receptors in the brain, and their functions are different. Thus, some work has suggested that 5-HT is a kind of love neurotransmitter, while others have suggested that it is a kind of disgust neurotransmitter [16,64,88]. Serotonin, a similar kind of chemical in the blood and gut, is closely related to disgust, nausea and vomiting [14]. Indeed, 5-HT corresponds to the emotions of disgust and sadness, and introverts are prone to disgust and even depression. It has been hypothesized that introverts might be particularly vulnerable to frustrated non-reward effects and may be likely to develop reactive depression [89]. Recently, it was found that introverts are highly sensitive to punishment and punishment warnings [45]. Some studies have suggested that introverted personality traits have some connection with 5-HT. The harm avoidance response to aversive stimuli is regulated by the long and short alleles of the 5-HT promoter transporter, and it is the introverted personality that corresponds to the dimensions of disgust and harm avoidance [90]. Thus, 5-HT might be the neurotransmitter associated with aversive processing [91]. However, despite the fact that there has not been a sufficient amount of research exploring the relationship between 5-HT and introverted personalities, future research could focus on the neurotransmitter angle and combine personality traits to intervene in symptoms of anxiety and depression.

### 5.3. Neuroticism—Norepinephrine (NE)

Norepinephrine (NE), a type of catecholamine, is an important neurotransmitter in the sympathetic nervous system and is a synonym for stress. Most previous studies have investigated the link between NE and stress based on the physiological theory of personality proposed by Cloninger [9], which suggests that NE is associated with stressful mechanisms and its major function is “fight-or-flight” [92]. In our previous studies, we suggested that NE is the neurotransmitter for fear and anger [93,94]. We also suggested that fear and anger might be the same emotion, because they share the same neurotransmitter (NE). Indeed, fear and anger are interchangeable by the appraisal of the resources, and both result in fight-or-flight behaviors. We argue that future research should focus on the association between NE and neuroticism; neurotic individuals have a low activation threshold and are prone to experience negative emotions in the face of stress, which is known to activate the release of NE in the brain [93,95].

The diagram displayed in Figure 2 shows the correspondence between personality traits and emotions and personality traits and the neurotransmitters mentioned above.

## 6. Anxious Personality Traits

An affective trait is an emotional tendency or characteristic, sometimes referred to as temperament, that is individually stable and can be seen across contexts, situations and time [97,98,99]. The tendency of individuals to experience anxiety is different, and this individual difference is known as trait anxiety. People with high levels of trait anxiety are more likely to experience elevations in state anxiety [100]. Research has indicated that individuals with high emotional reactivity (high neuroticism) and introverted tendencies (low extroversion) are more likely to experience anxiety than other personality types [101]. Anxiety personality traits are related to the two dimensions of Eysenck’s personality theory, and their correlation is described below.

### 6.1. Anxiety—Neuroticism

Neuroticism is significantly correlated with anxiety disorders [23]. Some findings have suggested that people with high neuroticism scores may be more likely to feel anxious than those with low scores [43]. It has also been proposed that all anxiety disorders are related to neuroticism [102,103]. Personality dimensions, such as neuroticism, can have an impact on how a person copes with stress, and these effects have corresponding emotional consequences. Effective coping may relieve stress, while ineffective coping may lead to anxiety [55]. Lakuta recruited 135 subjects aged 18–50 years to investigate how the Big Five personality factors predicted social anxiety (SA) symptoms [104]. Personality traits were measured, and SA symptoms were assessed one month later. The results showed that low emotional stability (neuroticism) was an independent predictor of higher SA levels. In Goldberg’s Big Five questionnaire, there is a “worried” item (emotional stability), suggesting a link between anxiety traits and neuroticism [105]. In their study of the relationship between personality and psychopathology, Middeldorp found that neuroticism was highly correlated with anxiety and depression [42]. High neuroticism in youth is a true risk factor or risk marker for first-onset anxiety and depressive disorders [106].

### 6.2. Anxiety—Introversion

In addition to neuroticism, anxiety traits are also correlated with introversion; however, their correlation is smaller than that between neuroticism and anxiety, roughly in a ratio of 1:2, as reflected in the model personality trait–emotion–neurotransmitter diagram (Figure 2) [89]. Analyses have also shown that after controlling for the effects of the other dimensions of the “Big Five”, only neuroticism and extroversion remain significant independent predictors of social anxiety in patients [107]. However, only one-third of social anxiety patients fit the “anxious-introverted” (shyness) personality profile associated with the disorder, which also suggests that although introversion and anxiety are associated, the correlation is not very strong. According to Grimm, increased glutamate concentration in the dorsal lateral prefrontal cortex (DLPFC) is negatively associated with the anxiety state, which might be a neural mechanism specific to introverted individuals and anxiety traits [108]. Gruda and Ojo examined the longitudinal relationship between extroversion and state anxiety in a large population of New York City (NYC) residents using a linguistic analysis machine learning approach [109]. In normal life, extroverts experience lower state anxiety than introverted individuals. However, this difference narrowed between extroverted and introverted individuals after the SARS-CoV-2 epidemic; thus, the associations between extroversion and positive mood and introversion and negative mood need to be re-examined in this larger context.

Overall, anxiety traits are correlated with neuroticism and introversion but have a greater association with neuroticism. People with high neuroticism and introversion scores are more likely to feel anxious.

### 6.3. Emotional Manifestations of Anxiety Personality Traits

“The oldest and most intense human emotion is fear, and the oldest and most intense fear is fear of the unknown” [79]. Research has shown that reducing fear by inhibiting the simulation of fearful events may be an effective coping strategy, suggesting that deficits in this coping style may contribute to the development of anxiety [110], as anxiety is the fear of the future. Fear is usually an adaptive emotion that allows us to react to immediate threats and helps us to anticipate future dangerous situations; however, it becomes pathological if the fear is excessive or irrational and severely affects one’s daily life. Thus, anxiety is excessive or irrational fear [111]. According to the DSM-V, anxiety disorders have “common features of excessive fear and anxiety related behavioral disturbances” [112,113]. Although the symptoms of fear and anxiety are similar, there still are some differences, for example, anxiety is a fear that points to the future, whereas fear refers more to the emotional state of the present.

Anxiety traits are characterized by emotional engagement with negative information. However, the related basic emotions seem to be overlooked when studying anxiety, for example, disgust, which was shown to be one of the basic emotions that is closely related to anxiety traits [14]. A broad range of theories from psychological disciplines consistently suggest that uncertainty is inherently perceived as disgust [79], and that uncertainty can often induce anxious emotion. Aversion sensitivity has indeed been found to be associated with generalized anxiety disorders [114]. The purpose of this study was to investigate the relationship between aversion sensitivity and trait anxiety or anxiety disorders. The results showed that aversion sensitivity is positively associated with generalized anxiety disorders and trait anxiety. Several other studies have also shown that the experience of aversion is associated with the development of specific anxiety disorders [115], for example, Woody and Teachman suggested that disgust plays a functional role in some anxiety disorders, particularly in specific phobias [74].

Fear usually stimulates avoidance to reduce the sense of danger, and avoidance is a central part of the problems associated with anxiety disorders. It is likely that disgust plays a similar protective role [74]; thus, the two emotions, fear and disgust, are linked with anxiety traits. When treating anxiety-related diseases in the future, we can intervene with these two emotions of fear and disgust, which has high clinical significance.

### 6.4. Neuro-Humoral Mechanisms of Anxiety Personality Traits

Establishing a relationship between neurotransmitters and behavioral responses is difficult because different stimuli (electrical, chemical and behavioral) elicit a range of different neurotransmitter releases, while at the same time, different neurotransmitters can elicit the same response and one neurotransmitter can elicit different responses [116]. Still, previous research has established a relationship between anxiety traits and certain neurotransmitters, such as norepinephrine (NE), 5-hydroxytryptamin (5-HT), and GABA (gamma-aminobutyric acid), which have been shown to be associated with the development and relief of anxiety [116]. High trait anxiety may be associated with hormonal release from acute stressful stimuli [117]. Although previous studies have demonstrated the influence of several neurotransmitters on anxiety, the brain region-specific neural mechanisms that contribute to individual differences in anxiety traits remain unclear. There is a great distinction between anxiety traits as an enduring personality trait and anxiety state, and the distinction between them is critical to anxiety research [118].

## 7. Discussion

### 7.1. Conclusions

The theory of personality traits has been continuously developed, and a large number of studies have been conducted to analyze mental disorders from the perspective of personality traits. In this paper, we analyzed personality traits from the perspectives of the basic emotion model and neurotransmitters by comparing with the two personality traits of extroversion and neuroticism proposed by Eysenck, placing special emphasis on anxiety personality traits, which are important for the clinical research of related mental disorders.

This paper put forward the following correlations (personality trait–emotion–neurotransmitter): extroverted–happy–dopamine (DA); introverted–disgust–5-hydroxytryptamine (5-HT); unstable–anger/fear–noradrenaline (NE); and stable–calmness. An extroverted personality is more associated with pleasure emotions, and extroverts often feel positive emotions such as joy because all the things that satisfy our needs are outward. Extroverted people are happier than introverted people. It has also been shown that extroversion is related to the dopaminergic system. Introverted personality traits are more prone to aversive emotions, and people who are too egoistic and introverted are prone to aversion, or depression. Neurotic individuals, i.e., those with emotionally unstable personality traits, are prone to encounter “fight or flight” scenarios, which could induce anger or fear emotions and are regulated by NE; in contrast, those with emotionally stable personality traits are more likely to feel relaxed and calm emotions. In this paper, the specific relationship between the personality trait of emotional stability, emotion (calmness/relaxed) and neurotransmitter (ACh) was not reviewed for the time being because there are few studies in this area; and future studies should be conducted to verify the exploration of this dimension.

In the model of personality traits–emotions–neurotransmitters, the anxious personality trait was specially analyzed. According to previous studies, the correlation between anxiety and neuroticism is two times the correlation between anxiety and introverted personality traits. In this paper, we examined the relationship between personality traits, emotions and neurotransmitters related to anxious traits. From a personality trait perspective, individuals with high neuroticism scores were more likely to feel anxious than those with low scores, and introverted individuals experienced higher state anxiety than extroverted individuals, suggesting a correlation between anxiety traits, neuroticism and introversion. From the perspective of emotions, people with high anxiety traits tend to feel fear and arousal. From the perspective of neurotransmitters, anxiety personality traits are indeed regulated by 5-HT, NE, GABA and HPA, in addition to the neurotransmitters that play a common regulatory role, with the regulation of 5-HT being more prominent.

### 7.2. Expectations

There are limitations in the current research on the neuro-humoral mechanisms of anxiety personality traits. For example, there are few studies that distinguish between state anxiety and trait anxiety from the perspective of neurotransmitters. State anxiety is a result of the interaction of individual trait anxiety with context [119,120] and is found in the presence of contextual conditioning. Low anxiety traits may produce severe state anxiety as much as high anxiety traits [116]; hence, there may be differences in the neuro-humoral mechanisms of state anxiety and trait anxiety. Second, there is a growing body of research suggesting that personality changes over the course of life [121]; thus, the effects of deviations in neurotransmitters on personality changes is a direction that could be investigated in the future.

Personality traits strongly influence quality of life [122], and empirical studies of personality and psychopathology have led to a growing recognition that many psychiatric disorders represent maladaptive variants of normative personality traits. Personality traits may increase the risk of developing psychiatric disorders, and exploring personality traits in relation to psychiatric disorders is of great relevance. In this paper, we explored the neuro-humoral mechanisms of anxiety personality traits. Future research could discuss the neural mechanisms of anxiety personality traits, such as brain imaging mechanisms and electroencephalographic mechanisms. It would be extremely helpful for the clinical treatment of personality disorders if interventions for personality disorders could be performed through the modulation of neurotransmitters and emotions. In addition, future research could focus on the emotional expression and neuro-humoral mechanism of anxiety personality traits and establish a systematic personality–emotion–neurotransmitter model, which may have important implications for the treatment of anxiety disorders.

## Figures and Tables

**Figure 1 brainsci-12-01141-f001:**
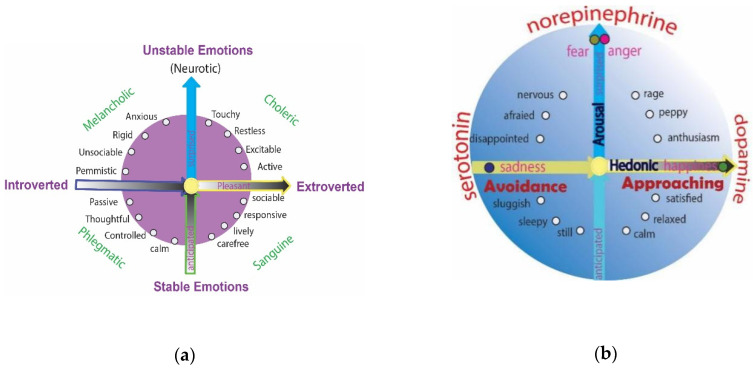
Similarity between Eysenck’s dimension and emotional dimensions. (**a**) Eysenck described two factors to account for variations in our personalities: extroversion/introversion and emotional stability/instability [34]. (**b**) Dimensional theories about emotions and basic emotions with monoamine neurotransmitters [38].

**Figure 2 brainsci-12-01141-f002:**
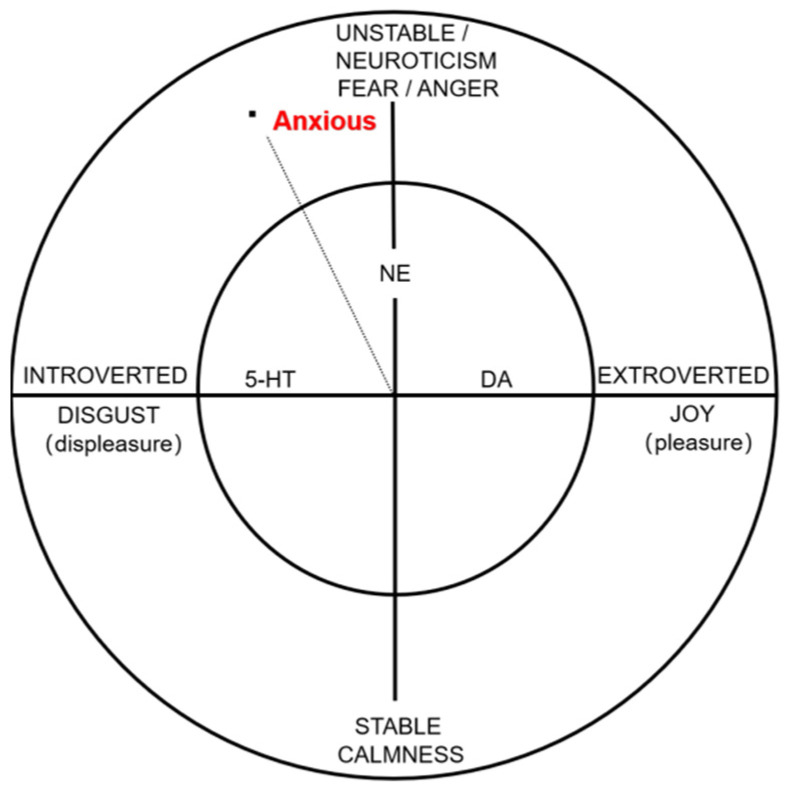
Personality trait–Emotion–Neurotransmitter diagrammatic figure. Extroverted–joy–dopamine (DA); introverted–disgust–5-hydroxytryptamine (5-HT); unstable (neuroticism)–anger/fear–noradrenaline (NE); stable–calmness. Anxiety traits are associated with both neuroticism and introversion, and the ratio of correlations is approximately 2:1. Anxiety traits are associated with emotions such as fear (anger) and disgust and are regulated by 5-HT and NE [16,89,96].

## Data Availability

The authors declare that all documents are available upon request.

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
