# Peer review of "Anxious Personality Traits: Perspectives from Basic Emotions and Neurotransmitters"

_brainsci, 2022, doi:10.3390/brainsci12091141_

Round 1

Reviewer 1 Report

This is a very interesting paper reviewing basic emotions and neurotransmitters associated with anxious personality traits. The paper is well-written and is of interest for the readers and the journal; however, several minor changes should be made.

Abstract.

1- I recommend to divide the abstract into several sections. Which methods did the authors use to conduct the review?

2- It is difficult to separate, in the abstract, the results and the conclusions. 

Introduction.

1- The introduction is really short. I recommend to expand the last part on personality traits.

2- At the end of the introduction, should the authors report the main aims of this review. Which questions are the authors addressing?

Results

1- The main sections of the review shoudl be divided according to the questions that the authors are addressing.

For instance, section 3. "The similarity between Eysenic's dimension and emotional dimensions" should be described according to the hypothesis the authors are testing.

2- Neutransmitters for personality traits.

Discussion. 

1- The authors are describing conclusions instead of Discussion. I recommend to rename it as Discussion.

2- Summary is not a good section for the discussion, as a summary is not to be provided in the discussion or in the conclusion. 

3- Which are the main conclusions related with anxious traits and emotions, and neurotransmitters? These conclusions should be developed more in depth. I recommend to add a couple of paragraphs focused on Future perspectives.

Reviewer 2 Report

I have read with great interest the manuscript entitled: Anxious personality traits: perspectives from basic emotions and neurotransmitters. In this study, Dong et al described personality traits from the perspective of neurotransmitters, while focusing on the emotional manifestations and neurohumoral mechanisms of anxiety-related personality traits in a systematic way. This will provide a new understanding and novel perspective for the study of neurotransmitters and personality traits, as well as an important reference for the development of neuropharmaceuticals. Authors tried to compare recent emotional studies with personality traits to systematically and comprehensively explore personality traits from the perspective of emotion feeling, and try to relate them with chemical in the body. This paper try to connect personality with chemicals in the body by putting forward the following correspondence (personality trait - emotion feeling - neurotransmitter).

The manuscript is of good impact, well conceptualized, and clearly written. The introduction provide sufficient background and include all relevant references.The conclusions supported by the evidence presented. All the cited references are relevant to the research. I think that the topic of this manuscript is important and is suitable for Brain Sciences (a special issue: The Neural Base of Personality and Adulthood Behavioral Disorders).

Morever:

Authors should describe acknowledgments.

Lines 53 “Although many studies have tried to sort up the physiological factors of personality traits, there is not a clear theory about personality, or a direct link between neurotransmitters and personality traits” - Authors should add  references.

Thank you
